# Mechanisms of Antitumor Invasion and Metastasis of the Marine Fungal Derivative Epi-Aszonalenin A in HT1080 Cells

**DOI:** 10.3390/md21030156

**Published:** 2023-02-26

**Authors:** Yi Liu, Liyuan Lin, Haiyan Zheng, Yuan-Lin He, Yanmei Li, Chunxia Zhou, Pengzhi Hong, Shengli Sun, Yi Zhang, Zhong-Ji Qian

**Affiliations:** 1School of Chemistry and Environment, College of Food Science and Technology, Guangdong Ocean University, Zhanjiang 524088, China; 2Shenzhen Institute of Guangdong Ocean University, Guangdong Ocean University, Shenzhen 518108, China; 3Collaborative Innovation Center of Seafood Deep Processing, Dalian Polytechnic University, Dalian 116034, China

**Keywords:** alkaloids, neoplasm metastasis, matrix metalloproteinases, cell signaling

## Abstract

Epi-aszonalenin A (EAA) is an alkaloid that is isolated and purified from the secondary metabolites of coral symbiotic fungi and has been shown to have good atherosclerotic intervention activity and anti-angiogenic activity in our previous studies. In the present study, antiangiogenic activity was used as a basis of an intensive study of its mechanism of action against tumor metastasis and invasion. Invasive metastatic pairs are a hallmark of malignancy, and the dissemination of tumor cells is the most dangerous process in the development of tumors. The results of cell wound healing and the Transwell chamber assay showed that EAA interfered well with PMA-induced migration and invasion of HT1080 cells. Western blot and the ELISA assay showed that EAA decreased MMPs and vascular endothelial growth factor (VEGF) activity and inhibited the expression of N-cadherin and hypoxia-inducible factor-1α (HIF-1α) by regulating the phosphorylation of downstream mitogen-activated protein kinase (MAPK), PI3K/AKT, and NF-κB pathways. Simultaneous molecular docking results revealed that the mimic coupling between the EAA and MMP-2/-9 molecules formed a stable interaction. The results of this study provide a research basis for the inhibition of tumor metastasis by EAA, and together with previous studies, confirm the potential pharmacology and drug potential for this class of compound for application in angiogenesis-related diseases and further improve the availability of coral symbiotic fungi.

## 1. Introduction

Metastasis is one of the most common and fatal causes in the treatment of cancer patients. In addition, significant signs of malignant tumors also include tumor metastasis and invasion. In recent decades, advanced cancers were mostly diagnosed with cytotoxic chemotherapy, and resistance to chemotherapy continued to develop and tumors continued to grow even after the recovery period [1]. In contrast, protein-based targeted therapies are becoming more prominent in clinical performance and can reduce the side effects of anticancer drugs [2]. Infinitely increasing tumor cells are highly invasive and metastatic to the extent that they spread to other organs and tissues without complete removal, ultimately leading to the necrosis of body functions, which is the main reason why tumors are difficult to cure [3]. Angiogenesis is one of the key factors in tumor metastasis. Tumor metastasis includes epithelial mesenchymal transformation (EMT), migration, and the invasion of tumors. Invasion is the first step in tumor metastasis. The expression of adhesion molecules on the surface of tumor cells is reduced, and cells become more motile and can break the basement membrane of endothelial cells in blood vessels or lymphatic vessels to enter the circulatory system and move to other organs or tissues of the body with flow of the lymph fluid and blood [4]. Tumor metastasis and growth depend on large amounts of nutrients and oxygen for energy, and when cells cannot take up sufficient energy sources, this can leave tumor cells in an environment with a low level of oxygen and nutrients, a situation that promotes the production of hypoxia-inducible factors (HIF-1α) and vascular endothelial growth factor (VEGF), among others, to provide a pathway for tumor cells to obtain blood through induced angiogenesis, providing sufficient nutrients and oxygen to continue their growth [5].

Matrix metalloproteinase (MMPs), a class of proteases with hydrolytic properties, play an important role in the degradation of the extracellular matrix (ECM) to accelerate tumor metastasis as well as in promoting the ability of cells to invade and metastasis and intertumor vascularization during the invasion and metastasis of tumor cells to various parts of the body [6]. Among these, MMP-2 is expressed most frequently when tumor cells undergo invasive behavior, while MMP-9 is expressed most frequently when tumor cells undergo metastatic behavior. MMP-9 also plays an important role in promoting angiogenesis within malignant tumors by regulating the expression of VEGF and promoting the specific binding of VEGF to its receptor to activate angiogenesis as well as by degrading the extracellular matrix around the vascular endothelial cells to allow endothelial cells to sprout and form intertumor neovascularization [7]. 

As a potential source of natural products, in recent years, more and more active substances have been found to have antitumor activity. Among them, marine microorganisms such as marine fungi and bacteria are valuable sources of active natural products and have been proven to have antibacterial, anti-inflammatory, antitumor, and antiviral biological activities [8,9,10]. Epi-aszonalenin A (EAA) is an alkaloid isolated from a secondary metabolite of the marine coral endophytic fungus *Aspergillus terreus* C23-3 (Figure 1A). Our previous study showed that it exerts a modulatory effect on atherosclerotic lesions and can have inhibitory activity on intraplaque angiogenesis [11]. This led us to further investigate whether EAA has antitumor activity. The HT1080 cell line has high metastatic properties and is a classical model of tumor cell metastasis. Phorbol-12-myristate-13-acetate (PMA) can act as a pro-cancer factor by activating PKC [12], and is therefore often used as an effective tumor promoter. In this study, we used the PMA-induced HT1080 cell model in order to confirm the effect of EAA on tumor migration and invasion, and to analyze some of the mechanisms. The effect of EAA on the metastatic invasive ability of HT1080 cells was first examined by scratch and invasion assays. Molecular docking was used to simulate the intermolecular forces between EAA and MMP-2/-9. Second, Western blot, ELISA, and immunofluorescence were used to detect the effects of EAA on HT1080 cell-associated proteins. This is beneficial to better understand the metabolites of coral endophytes and provide an experimental basis for the development of EAA into functional foods and pharmaceuticals.

## 2. Results

### 2.1. Effect of EAA on the Viability of HT1080 Cells

The results of the cell viability assay showed that EAA was not toxic to the HT1080 cells at any concentration on 20 μM, and the difference in cell viability between the groups was not statistically significant (Figure 1B). Therefore, we can conclude that EAA (0.1–20 μM) is not cytotoxic and can be used in subsequent experimental studies.

### 2.2. Effect of EAA on Migration Capacity and Invasion of HT1080 Cells

PMA is known to significantly enhance the invasion and migration of HT1080 cells [13]. In the Transwell cell invasion assay, the highest number of cells was observed in the control group, while the number of cells was greatly reduced in the EAA-treated group (Figure 1C). In addition, in the wound healing assay, the control group cells rapidly migrated to the wound site, whereas the migration ability of HT1080 cells was significantly reduced after treatment with EAA (Figure 1D). The results of both experiments were proportional to the concentration, with the highest inhibition rate achieved at 10 μM. These results suggest that EAA has anti-tumor invasive metastatic activity at the cellular level and deserves further study.

### 2.3. Effect of EAA on HIF-1α, VEGF, and N-Cadherin Expression in the HT1080 Cells Induced by PMA

In the present study, the expression of HIF-1α, VEGF, and N-cadherin was measured by protein blotting (Figure 2A), and the results showed that PMA could activate the overexpression of these three proteins while EAA could exert significant inhibitory effects on their expression, and the concentration was proportional to the inhibitory effect. Therefore, it can be concluded that EAA can effectively inhibit the protein expression of HIF-1α, VEGF, and N-cadherin in the HT1080 cells, thus achieving the effect of tumor suppression.

### 2.4. Effect of EAA on IL-1β and IL-6 Expression in HT1080 Cells Induced by PMA

IL-1β and IL-6 are cellular inflammatory components, and these factors modify the perivascular endothelial cell surroundings and promote tumor development [14]. In this research, we found that inflammatory factors IL-1β and IL-6 were significantly downregulated in HT1080 cell supernatants under the influence of EAA (Figure 2B). Therefore, it can be concluded that EAA inhibits the protein expression of inflammatory factors of HT1080.

### 2.5. Effect of EAA on MAPK and PI3K/Akt Pathway in PMA-Induced HT1080 Cells

MAPK and PI3K/AKT signaling are associated with many malignancies [15]. Protein immunoblotting was used to detect the PMA-induced expression of the MAPK and PI3K/AKT signaling pathways in the HT1080 cells. However, treatment with EAA (0.1, 1 and 10 µM) effectively inhibited their phosphorylation to suppress the activation of the MAPK and PI3K/AKT signaling pathways (Figure 3).

### 2.6. Effects of EAA on PMA-Induced NF-κB Pathway and DNA-Binding Activity in HT1080 Cells

The NF-κB signaling pathway is closely associated with tumor development, and the activation and translocation of NF-κB trigger DNA transcription, which in turn leads to various inflammatory responses and apoptosis [16]. Figure 4A shows that PMA can activate the phosphorylated expression of IκB-α and p65 proteins, which are significantly inhibited by EAA. As shown in Figure 4B, PMA promotes the significant transfer of NF-κB p65 into the nucleus. However, EAA gradually decreased the DNA-binding activity of NF-κB. The results suggest that EAA can downregulate NF-κB expression and nuclear translocation, thereby inhibiting HT1080 cell proliferation and invasion.

### 2.7. Effect of EAA on MMPs Activity and Expression in HT1080 Cells Induced by PMA

The effects of EAA on the activity and expression of MMPs in the PMA-induced HT1080 cells were investigated by ELISA and protein blotting. The protein blotting results showed that the expression of MMPs was increased in the HT1080 cells treated with PMA, whereas the expression of MMPs (MMP-1/-2/-3/-9) was inhibited after treatment with EAA (Figure 5A). In addition, as shown in Figure 5B, the secretion of the MMP-2/-9 protein in the cell supernatant was significantly increased in the control group, and EAA was able to inhibit it in proportion to the concentration. The results indicate that EAA can effectively inhibit the protein activity of MMPs in the HT-1080 cells, thus achieving the effect of inhibiting tumor invasion and metastasis.

### 2.8. Molecular Simulation of the Effect of Docking EAA with MMP-2/-9

The interaction of EAA with the active pocket of the MMP-2/-9 protein was simulated by molecular docking, and the optimal docked 2D structure was obtained (Figure 6). Under this structure, the active pockets of MMP-2/-9 and the alkaloid EAA showed a compact binding pattern (Figure 6A,C). As shown in Figure 6B,D, EAA can form three hydrogen bonds with THR-145, GLY-135, and ALA-136 on the MMP-2 protein and two hydrogen bonds with GLN-227 and TYR-248 on the MMP-9 protein. These hydrogen bonds allow the two to form a tight and stable complex.

## 3. Discussion

Tumor invasion metastasis and angiogenesis are the most critical processes that contribute to tumor growth. Many antitumor drugs cannot achieve ideal therapeutic effects due to the mixed use of multiple drugs, drug resistance, and other reasons. When one pathway is inhibited by drugs, other related pathways will compensate accordingly, resulting in a poor therapeutic effect. Therefore, multi-target therapy is considered to be the most effective approach to inhibit antitumor drug resistance and maximize antitumor effects [17]. An increasing number of studies has shown that the active ingredients of natural products overcome the drug resistance of tumor cells through multiple molecular mechanisms and have the effect of preferentially killing cancer molecular mechanisms with low side effects, resistance, and modulation [18,19,20], while several studies have illustrated the antitumor activity of alkaloids in the ocean [21]. EAA is a kind of indole alkaloid compound that is currently less studied and is widely distributed among nature, with various structural derivatives and strong pharmacological activity [22,23]. On the basis of previous studies, we demonstrated its ability to interfere with atherosclerotic plaque lesions. It is well-known that angiogenesis plays a crucial role in tumor lesions, and this study builds on the antiangiogenic activity demonstrated for the previous study to demonstrate its ability to combat tumor invasion and metastasis in a multitargeted manner.

Therefore, we first proved that EAA inhibits the migration and invasion of HT1080 cells based on noncytotoxicity, indicating that EAA has anti-cell metastasis and invasion activities (Figure 2). It has been shown that in hypoxic environments, HIF-1α is considered a key protein that can effectively regulate the production of cytokines related to invasive metastasis and angiogenesis, which are essential for tumor cells to obtain sufficient oxygen and nutrients [24]. VEGF promotes the growth of endothelial factors in arterial, venous, and lymphatic vessels. It can specifically affect blood vessels and promote angiogenesis during physiological and pathological processes [25]. N-cadherin is a marker of EMT [26] and can regulate the expression of MMP-9 [27]. The experimental results showed that EAA significantly decreased the expression of HIF-1α, VEGF, and N-cadherin proteins in the HT1080 cells, thus inhibiting tumor growth and achieving an inhibitory effect on tumor cell invasion and metastasis.

During infiltration and metastasis, weakly bound tumor cells must attach to the extracellular matrix and degrade it. MMPs degrade the basement membrane and ECM, promote endothelial cell migration, promote inflammatory responses, and regulate angiogenic factors to promote angiogenesis [28]. MMP-1/-2/-3/-9 are important tumor-associated proteins in the MMP family. MMP-2/-9 are gelatinizing, which is the key to basement membrane breakdown [29]. The experimental results showed that the expression of MMPs decreased significantly in response to 7PE and better at high concentrations, showing a concentration-dependent trend. We predicted the possible interaction between the EAA and MMP-2/-9 proteins using a molecular docking approach. The results showed that stable hydrogen bonds could be formed among them, which also provided a theoretical basis for the further mechanism of action of the inhibition of MMP-2/-9 of EAA.

The NF-κB, MAPK, and PI3K/AKT pathways are the key regulators of tumor cells [30]. Among them, NF-κB has the function of inhibiting apoptosis and is closely related to several processes such as tumorigenesis, growth, and metastasis [31]. NF-κB in the normal cytoplasm is inactivated and binds to the inhibitory protein IκB-α in a trimeric complex. The activation of NF-κB is followed by phosphorylation of the IκB-α protein, followed by nuclear ectopic of the p65 subunit, and regulation of the downstream signaling pathways and proteins [32], promoting the expression of MMPs, VEGF, and N-cadherin in the tumor cells [33,34]. Our results suggest that EAA inhibition of MMPs, VEGF, and N-cadherin proteins is associated with the inhibition of the NF-κB pathway.

Abnormal activation of the MAPK signaling pathway is closely associated with a variety of cancers and can cause tumorigenesis and growth through various mechanisms such as cell proliferation, stress, inflammation, differentiation, transformation, and apoptosis [35]. The PI3K/AKT pathway is closely associated with the tumorigenesis and growth of lung cancer, breast cancer, and endometrial cancer. It is not only a central pathway for protein translation that regulates tumor cell proliferation, growth, and survival, but is also a key step in tumor invasion and angiogenesis, and has been directly associated with drug resistance of treatment [36]. Therefore, it has been agreed by many scholars that the inhibition of its production may have an antitumor effect [37]. The results of the assay (Figure 4) showed that EAA effectively inhibited the PMA-induced phosphorylation of the MAPK and PI3K/AKT pathway proteins, thus suppressing tumor growth and metastasis.

## 4. Materials and Methods

### 4.1. Chemicals and Materials

EAA (Figure 1A) was purified from the secondary metabolite of the endophytic fungus *A. terreus* C23-3 secondary metabolite by silica gel vacuum liquid chromatography (VLC) elution and high performance liquid chromatography from coral *Pectinia paeonia*, which demonstrated its anti-inflammatory and anti-angiogenic activities in our previous study [11]. Human fibrosarcoma cells (HT1080) were purchased from the Cell Bank of the Chinese Academy of Sciences (Shanghai, China). Phorbol-12-myristate-13-acetate (PMA) was purchased from Sigma-Aldrich (St. Louis, MO, USA). Matrigel was purchased from BD Biosciences (San Jose, CA, USA). DAPI was purchased from Solarbio Science & Technology Co., Ltd. (Beijing, China). Dulbecco’s modified Eagle’s medium (DMEM), fetal bovine serum (FBS), and penicillin/streptomycin were products of Gibco (Grand Island, NY, USA). Primary mouse antibodies against VEGF (sc-7269), p65 (sc-8008), p-p65 (sc-136548), IκB-α (sc-1643), p-IκB-α (sc-8404), p38 (sc-535), p-p38 (sc-166182), ERK (sc-94), p-ERK (sc-81492), JNK (sc-7345), p-JNK (sc-6254), MMP-1 (sc-58377), MMP-3 (sc-21732), β-actin (sc-47778), and secondary antibodies goat anti-mouse IgG-HRP (sc-2005) and goat anti-rabbit IgG-HRP (sc-2004) were products of Santa Cruz Biotechnology (Santa Cruz, CA, USA). PI3K (#4257), p-PI3K (#17366), p70S6K (#2708), p-p70S6K (#9234), AKT (#4691), p-AKT (#4060), mTOR (#2972), p-mTOR (#5536), MMP-2 (#13132), and MMP-9 (#13667) were provided by Cell Signaling Technology (Boston, MA, USA). The ELISA kit was purchased from Elabscience Biotechnology Co., Ltd. (Wuhan, Hubei, China). Other chemicals used were analytical grade and are commercially available.

### 4.2. Cell Activity Assay (CCK-8)

HT1080 cells were grown in DMEM with 10% FBS and 1% penicillin/streptomycin. Cells were seeded in 96-well plates (1 × 10^4^ cells/well) and different concentrations of EAA were added to the wells for 24 h. A total of 10 μL of the Cell Counting Kit-8 (CCK-8) working solution was added and incubated for 1 h. The absorbance was measured using a microplate reader (BioTek, Winooski, VT, USA) to determine the OD value at 450 nm.

### 4.3. Cell Wound Healing Assay

HT1080 cells were seeded in 24-well plates (1 × 10^5^ cells/well) and cultured for 24 h at 100% density before using a sterile tip to scrape the cells. After washing away cellular debris, cells were treated with EAA for 1 h, and then stimulated with PMA (10 ng/mL). The migration of cells across the injury line was observed with a microscope (Olympus, Tokyo, Japan) and photographed at 6 h and 12 h, respectively.

### 4.4. Cell Invasion by the Transwell Chamber Assay

A layer of matrix gel (NEST Biotechnology, Wuxi, China) was placed on the upper layer of the cell compartment of the 24-well plate in advance and left to stand in the incubator for 30 min. Then, 200 μL of cell (1 × 10^5^ cells/well) suspension and EAA were added to the cell chambers and incubated for 24 h. Finally, the cells were stained, observed and photographed under a microscope (Olympus, Tokyo, Japan). 

### 4.5. Enzyme Linked Immunosorbent Assay (ELISA)

After the cells were treated with the samples and inducer, the cell supernatant was collected and the target protein concentration was tested according to the manufacturer’s instructions.

### 4.6. Western Blot

HT1080 cells were incubated with EAA (0.1, 1, and 10 μM) and PMA (10 ng/mL) in a CO_2_ incubator for 24 h by referring to the method of Chen et al. [38]. Proteins were obtained by lysing cells using RIPA buffer containing 1% of the protease inhibitor phenylmethylsulfonyl fluoride (PMSF). Separation was performed using SDS-PAGE. The proteins were then transferred onto nitrocellulose (NC) filter membranes. Primary antibodies were incubated overnight at 4 °C (1:1000), followed by sufficient incubation of secondary antibodies for 2 h (1:2000). Finally, the membranes were photographed using an enhanced chemiluminescence detection system (Syngene, Cambridge, UK), and the images were analyzed using ImageJ software.

### 4.7. Immunocytochemistry

Cells were treated as previously described (refer to the method of Pei et al. [39]). The cells were fixed with 4% paraformaldehyde for 1 h and permeabilized with 0.2% Triton X-100 for 10 min. Primary and secondary antibodies were incubated after blocking with 5% BSA, and the nuclei were stained with DAPI. Finally, images were recorded by inverted fluorescence microscopy (Olympus, Tokyo, Japan).

### 4.8. Molecular Docking

The structures of MMP-2 (PDB ID: 3AYU) and MMP-9 (PDB ID: 4HMA) were recovered from the Protein Data Bank. We used ChemBioDrawUltra 20.0 to draw the structure of the EAA compound and then converted it into a 3D structure using ChemBio3D Ultra14.0. Discovery Studio was used for the molecular docking studies [40]. The best EAA coupling sites at the protein active site (MMP-2/-9) were obtained separately.

### 4.9. Statistical Analysis

All results were expressed as the mean ± S.D (*n* = 3) and repeated three times. Statistical analysis was performed using Dunnett’s test (GraphPad Prism 5.0), one-way analysis of variance (ANOVA), and multiple comparisons. The plots were made with Origin 2021. The statistical significance of all tests was *p* < 0.05.

## 5. Conclusions

In conclusion, as shown in Figure 7, EAA has the ability to inhibit tumor invasion and metastasis, and combined with our previous findings, it was judged to have both antitumor and angiogenic activity. It could inhibit the PMA-induced expression of MMPs, key proteins for tumor invasion (HIF-1α and N-cadherin), and inflammatory factors (IL-1β and IL-6), and could inhibit angiogenesis by suppressing VEGF expression. Inhibition of the PI3K/AKT, MAPK, and NF-κB signaling pathways achieved the effect of inhibiting tumor invasion and metastasis. The results of the molecular docking also showed that EAA could form hydrogen bonds with MMP-2/-9 to bind and stabilize. The above findings suggest that EAA can effectively act on HT1080 with antitumor and anti-inflammatory effects. This study provides further theoretical guidance for the medicinal use of coral endophytes and research data for further experimental studies of EAA. In addition, EAA has the potential to be developed as a functional food ingredient or even as a drug that could be effective in treating angiogenesis-related diseases, but further in vivo animal studies and pharmacokinetics are needed to confirm its stability and assess specific tumor suppression effects.

## Figures and Tables

**Figure 1 marinedrugs-21-00156-f001:**
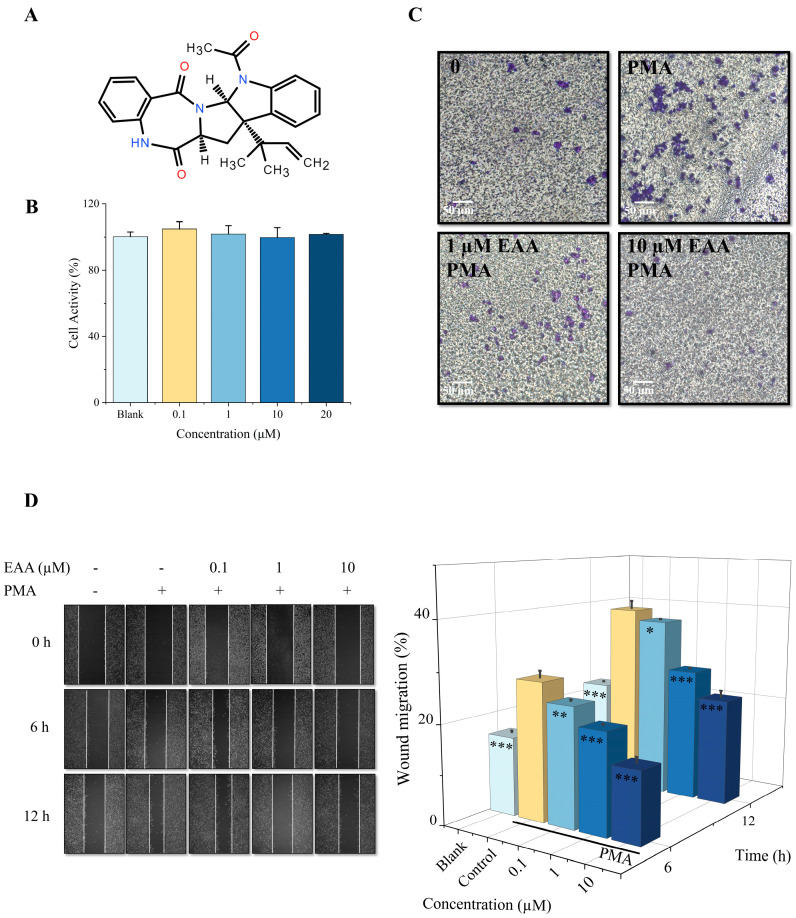
Effect of EAA on the viability of HT1080 cells. (**A**) Current status of the isolation study of active compounds in coral endophytic fungi and the chemical structure of the indole alkaloid Epi-aszonalenin A (EAA). (**B**) Effect of EAA on the cell viability of HT1080. (**C**) Inhibitory effect of EAA on the PMA-induced invasion of HT1080 cells. (**D**) Effect of EAA on the PMA-induced migration ability of HT1080 cells recorded at 0, 6, and 12 h. PMA was added at a concentration of 10 ng/mL. Data are shown as the mean standard deviation (*n* = 3). * Compared with the control group (PMA-treated cells), *p* < 0.05, ** Compared with the control group, *p* < 0.01, *** Compared with the control group, *p* < 0.001.

**Figure 2 marinedrugs-21-00156-f002:**
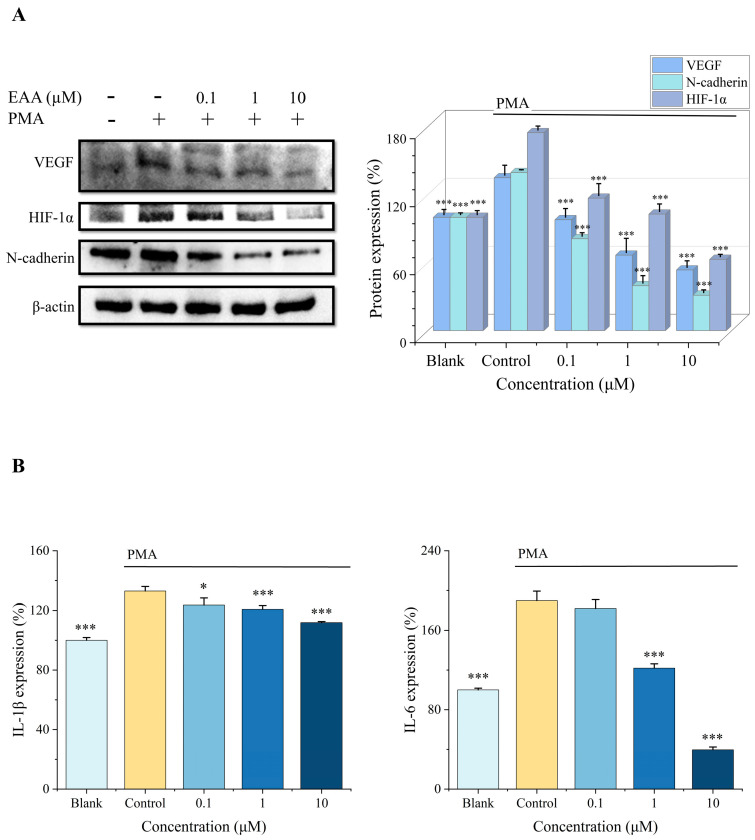
(**A**) Expression of VEGF, HIF-1α, and N-cadherin in the HT1080 cells was detected by Western blot. β-actin was used as an internal control. (**B**) Expression of IL-1β and IL-6 in the HT1080 cells was detected by ELISA. PMA was added at a concentration of 10 ng/mL. Data are shown as the mean standard deviation (*n* = 3). * Compared with the control group (PMA-treated cells), *p* < 0.05, *** Compared with the control group, *p* < 0.001.

**Figure 3 marinedrugs-21-00156-f003:**
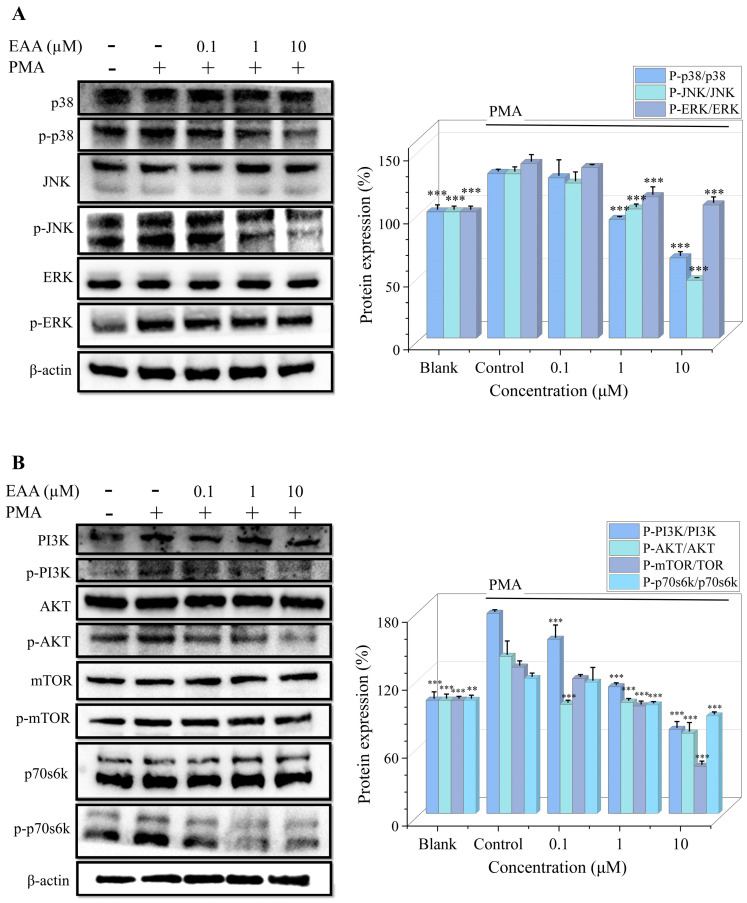
(**A**) Western blot detection of the protein expression levels of the MAPK signaling pathway in PMA-induced HT1080 cells. (**B**) Western blot detection of protein expression levels of the PI3K/AKT signaling pathway in the PMA-induced HT1080 cells. β-actin was used as an internal control. PMA was added at a concentration of 10 ng/mL. Data are shown as the mean standard deviation (*n* = 3). ** Compared with the control group (PMA-treated cells), *p* < 0.01, *** Compared with the control group, *p* < 0.001.

**Figure 4 marinedrugs-21-00156-f004:**
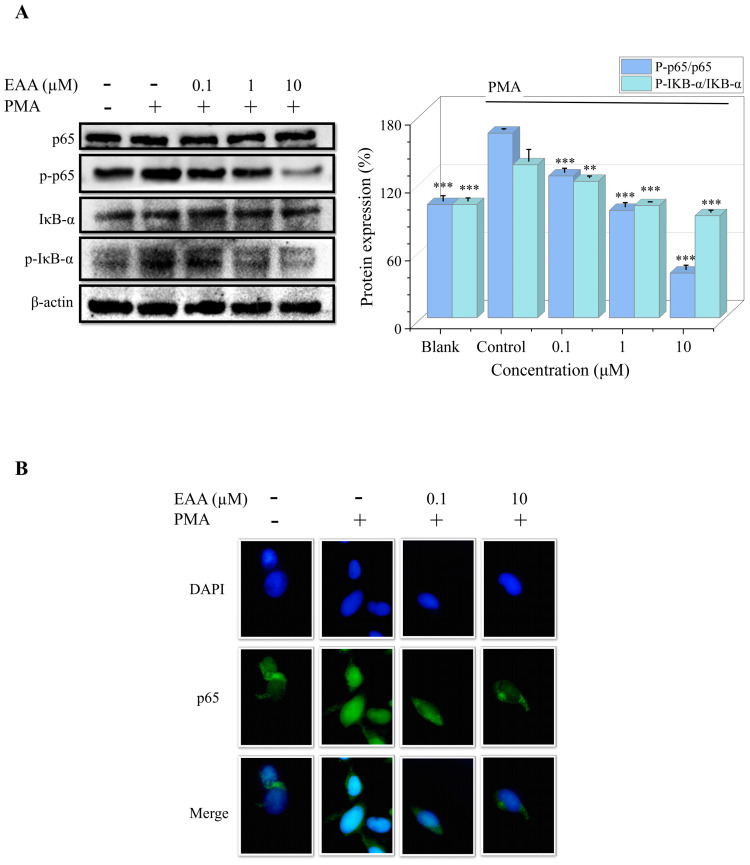
(**A**) Protein expression levels of the NF-κB signaling pathway in the PMA-induced HT1080 cells were determined by Western blot. β-actin was used as an internal control. PMA was added at a concentration of 10 ng/mL. (**B**) Nuclear translocation of p65 was observed by immunofluorescence through an overlay of blue DAPI staining with green p65 staining. Data are shown as the mean standard deviation (*n* = 3). ** Compared with the control group (PMA-treated cells), *p* < 0.01, *** Compared with the control group, *p* < 0.001.

**Figure 5 marinedrugs-21-00156-f005:**
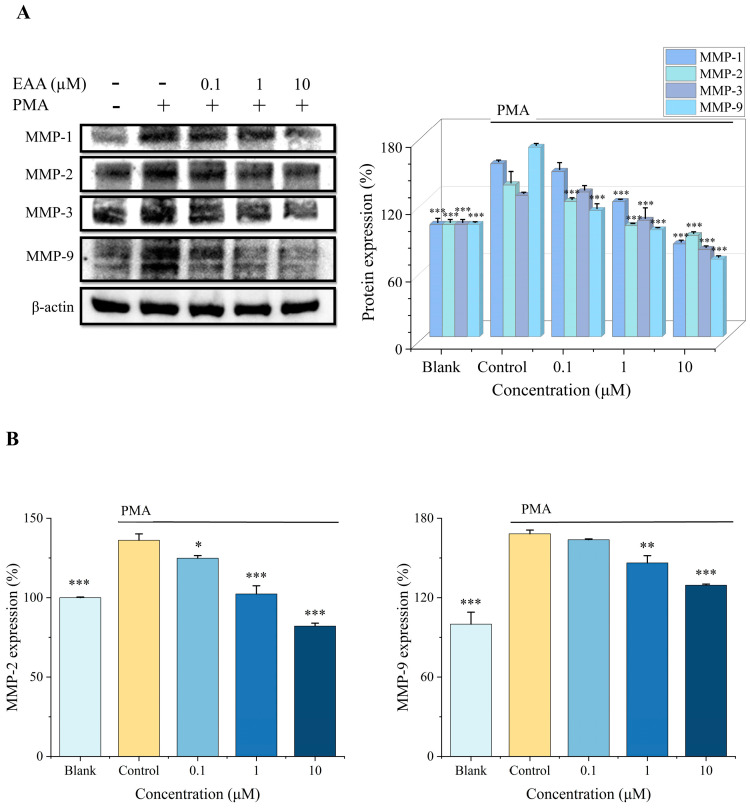
(**A**) The effect of EAA on the expression level of the MMP protein in the PMA-induced HT1080 cell lysates was detected by Western blot. β-actin was used as an internal control. PMA was added at a concentration of 10 ng/mL. Data are shown as the mean standard deviation (*n* = 3). (**B**) The effect of EAA on the secretion of the MMP-2/-9 protein in the HT1080 cell supernatant was measured by ELISA. * Compared with the control group (PMA-treated cells), *p* < 0.05, ** Compared with the control group, *p* < 0.01, *** Compared with the control group, *p* < 0.001.

**Figure 6 marinedrugs-21-00156-f006:**
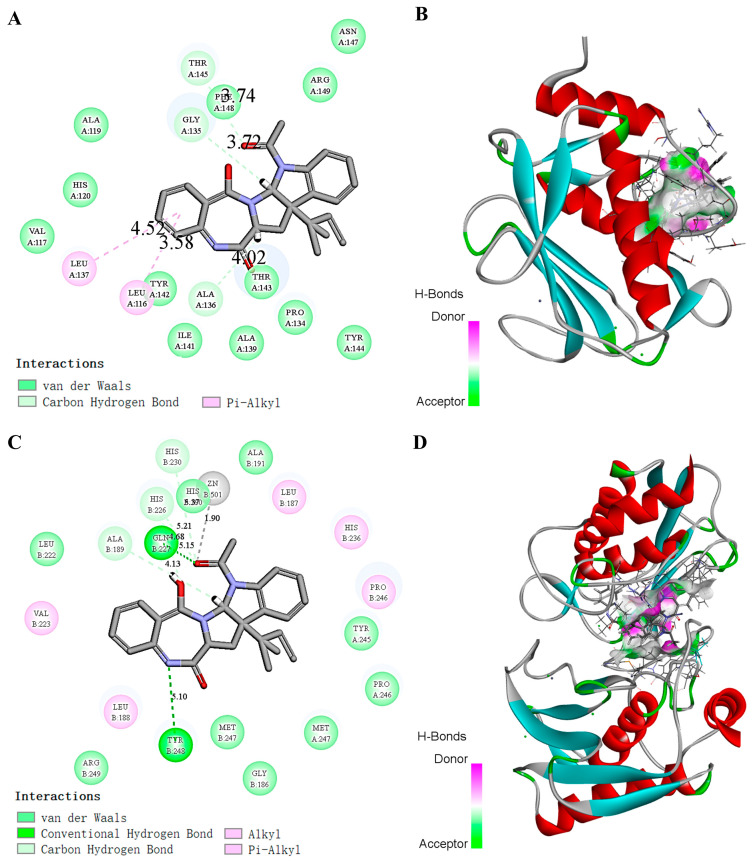
(**A**) Hydrogen bonding in the 2D molecular docking structure of EAA and the ligand MMP-2. (**B**) 3D molecular docking effect of EAA and the ligand MMP-2. (**C**) Hydrogen bonding in the 2D molecular docking structure of EAA and the ligand MMP-9. (**D**) 3D molecular docking effect of EAA and the ligand MMP-9.

**Figure 7 marinedrugs-21-00156-f007:**
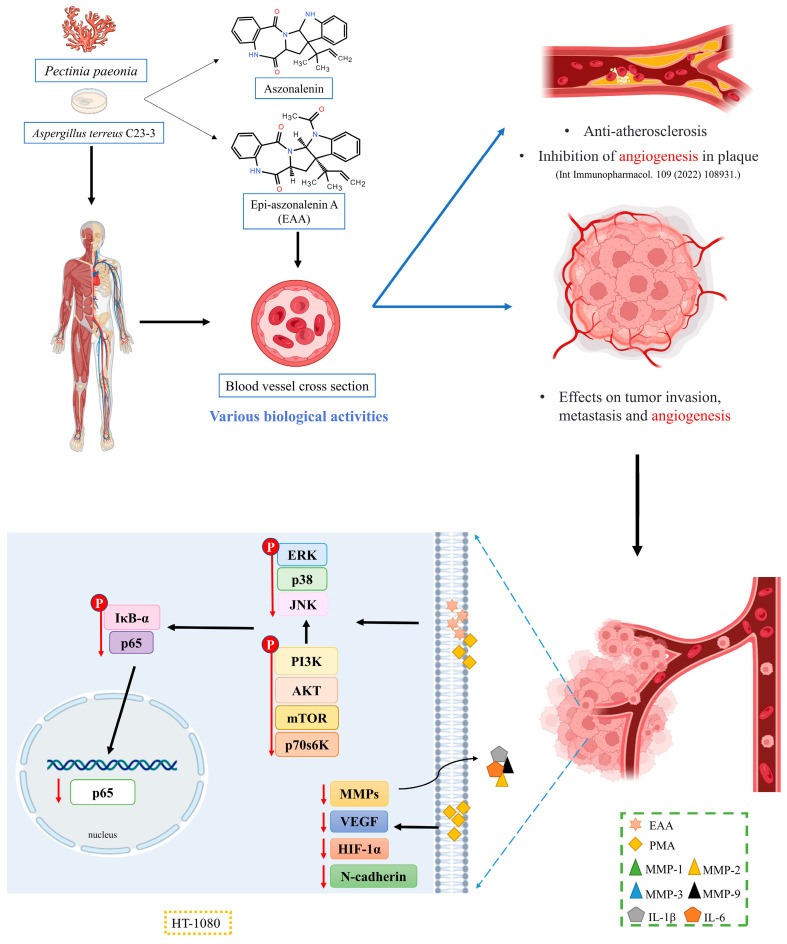
Main signal pathway of EAA inhibiting migration and invasion in HT1080 cells [11].

## Data Availability

All data generated or analyzed during this study are included in the manuscript.

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
