# Peer review of "Mechanisms of Antitumor Invasion and Metastasis of the Marine Fungal Derivative Epi-Aszonalenin A in HT1080 Cells"

_marinedrugs, 2023, doi:10.3390/md21030156_

Round 1
Reviewer 1 Report
The manuscript, “Mechanisms of antitumor invasion and metastasis of the marine fungal derivative EAA in HT1080 cells” by Liu et al. presents a study that suggests Epi-aszonalenin A (EAA) can effectively act on HT1080 with antitumor and anti-inflammatory effects. In conclusion, EAA has the potential to be developed into a functional food or even a drug that can act effectively in the treatment of angiogenesis-related diseases. Additionally, this study provides further theoretical guidance for the medicinal use of coral-derived endophytes and research data for further experimental studies on EAA.
Comments and Suggestions for Authors:
1- Lines 13-27 (Abstract): Explain about the material and method used in the study in the abstract section.
2- Line 28: Correct the keywords according to MeSH.
3- Line 99: Please add a scale bar for figure 1C.
4-Line 196-197: Please improve the quality of figure 6 so that the numbers are clear.
5-Lines 269-271: Please describe the purification method in this part and add the reference to the previous study.
6-Line 304 and 325: The sentence should not start with a number.
7-The article generally needs to be checked for grammatical errors.
Author Response
Response to Reviewer 1
Dear editor and reviewers:
On behalf of my co-authors, we thank you very much for giving us an opportunity to revise our manuscript and we appreciate editor and reviewers very much for your positive and constructive comments and suggestions on our manuscript entitled “Mechanisms of antitumor invasion and metastasis of the ma-rine fungal derivative EAA in HT1080 cells” (Manuscript Number: marinedrugs-2249015).
We have studied the reviewer’s comments carefully and have made revision in the manuscript. Point-by-point responses to the reviewers are listed below this letter. We have tried our best to revise our manuscript according to the comments.
Thank you for your kind comments and careful work.
The manuscript, “Mechanisms of antitumor invasion and metastasis of the marine fungal derivative EAA in HT1080 cells” by Liu et al. presents a study that suggests Epi-aszonalenin A (EAA) can effectively act on HT1080 with antitumor and anti-inflammatory effects. In conclusion, EAA has the potential to be developed into a functional food or even a drug that can act effectively in the treatment of angiogenesis-related diseases. Additionally, this study provides further theoretical guidance for the medicinal use of coral-derived endophytes and research data for further experimental studies on EAA.
Comments and Suggestions for Authors:
Comment 1: Lines 13-27 (Abstract): Explain about the material and method used in the study in the abstract section.
Response 1: Thanks for your comments. According to your suggestions, we have modified the abstract.
In Abstract section:
“The results showed that EAA had good intervention effects on PMA-induced migration and inva-sion of HT1080 and reduced the activity of MMPs and vascular endothelial growth factor (VEGF), inhibited nuclear ectopic of NF-κB p65 by regulating downstream mitogen-activated protein kinase (MAPK), PI3K/AKT and NF-κB pathway phosphorylation, inhibited N-cadherin and hypoxia inducible factor-1α (HIF-1α) expression. And simulated coupling interactions between EAA and MMP-2/-9 molecules were found to form stable interactions.” Corrected to“The results of cell wound healing and trans well chamber assay showed that EAA interfered well with PMA-induced migration and invasion of HT1080 cells. Western blot and ELISA assay showed that EAA decreased MMPs and vascular endothelial growth factor (VEGF) activity and inhibited the expression of N-adherin and hypoxia-inducible factor-1α (HIF-1α) by regulating the phos-phorylation of downstream mitogen-activated protein kinase (MAPK), PI3K/AKT and NF-κB pathways. Simultaneous molecular docking results revealed that the mimic coupling between EAA and MMP-2/-9 molecules formed a stable interaction.”
Comment 2: Line 28: Correct the keywords according to MeSH.
Response 2: Thanks for your careful work. According to your comments, we searched the MeSH terms and corrected the keywords in the revised draft.
In Keywords section:
“Coral endophytic fungus; Metastasis; MMPs; Signal pathway” Corrected to “Alkaloids; Neoplasm Metastasis; Matrix Metalloproteinases; Cell Signaling”
Comment 3: Line 99: Please add a scale bar for figure 1C.
Response 3: Thanks for your careful work. According to your comments, the related content has been corrected in our revised manuscript.
Figure section Corrected to below figure
Comment 4: Line 196-197: Please improve the quality of figure 6 so that the numbers are clear.
Response 4: Thanks for your careful work. According to your comments, we increased the resolution of figure6 and enlarged it, and the related content has been corrected in our revised manuscript, as shown below.
Figure section Corrected to below figure
Comment 5: Lines 269-271: Please describe the purification method in this part and add the reference to the previous study.
Response 5: Thanks for your careful work. As the extraction and isolation methods have been detailed in previous research articles, the main methods and references have been added here.
In 4. Materials and Methods section:
“EAA (Figure 1A) was purified from the secondary metabolite of the endophytic fungus A. terreus C23-3 secondary metabolite from coral Pectinia paeonia and demon-strated its anti-inflammatory and anti-angiogenic activities in our previous study.” Corrected to “EAA (Figure 1A) was purified from the secondary metabolite of the endophytic fungus A. terreus C23-3 secondary metabolite by silica gels vacuum liquid chromatog-raphy (VLC) elution and high performance liquid chromatography from coral Pectinia paeonia, and demonstrated its anti-inflammatory and anti-angiogenic activities in our previous study [11].”
Comment 6: Line 304 and 325: The sentence should not start with a number.
Response 6: Thank you for your kind comments and careful work. According to your comments, the related content has been corrected in our revised manuscript.
In 4. Materials and Methods section:
“100 μL of matrix gel (NEST Biotechnology, Wuxi, China) was added to the cell chamber and left to stand in the incubator for 30 min.” Corrected to “A layer of matrix gel (NEST Biotechnology, Wuxi, China) was placed on the upper layer of the cell compartment of the 24-well plate in advance and left to stand in the incubator for 30 min.”
“24 h were fixed with 4% paraformaldehyde for 1 h and permeabilized with 0.2% Tri-ton X-100 for 10 min” Corrected to “The cells were fixed with 4% paraformaldehyde for 1 h and permeabilized with 0.2% Triton X-100 for 10 min.”
Comment 7: The article generally needs to be checked for grammatical errors.
Response 7: Thanks for your suggestions. We feel sorry for our poor writings. We have been revised in a native English speaker help to polish our article. And we hope the revised manuscript could be acceptable for you.
We would like to express our great appreciation to you for comments on our paper.
Thank you and best regards.
Yours sincerely
Zhong-Ji Qian, Ph.D.
Professor
Southern Marine Science and Engineering Guangdong Laboratory, Zhanjiang, China
Shenzhen Institute of Guangdong Ocean University, Shenzhen 518108, China
School of Chemistry and Environment, Guangdong Ocean University, Zhanjiang 524088, China
Tel: +88-759-2396270
E-mail: zjqian78@163.com

Reviewer 2 Report
Authors have presented the Mechanisms of antitumor invasion and metastasis of the marine fungal derivative EAA in HT1080 cells. The concept of MS is good. However, the manuscript can be improved by improving experimental section.
The specific comments, which could help to improve the manuscript are:
Manuscript should be revised for grammatical & punctuation errors.
Better to Provide Full form of EAA in the Title.
Page 7; line 195: How Authors are sure about the stability of complex without performing MD simulation studies?
Page 9, line 269-270: EAA (Figure 1A) was purified from the secondary metabolite of the endophytic fungus A. terreus C23-3 secondary metabolite from coral Pectinia Paeonia. Provide a suitable reference.
If authors referred earlier related publications, provide a suitable reference for Section 4.2 to 4.7.
Page 12; line 356: On what basis authors claim “EAA has the potential to be developed into a functional food?
Authors mentioned in Abstract that “EAA had good intervention effects on PMA-induced migration and invasion of HT1080 and reduced the activity of MMPs and vascular endothelial growth factor (VEGF). However, Molecular Docking Studies were performed only on MMP-2/-9.
It would be better to add research gap and future prospects related to the topic in conclusion section.
Author Response
Response to Reviewer 2
Dear editor and reviewers:
On behalf of my co-authors, we thank you very much for giving us an opportunity to revise our manuscript and we appreciate editor and reviewers very much for your positive and constructive comments and suggestions on our manuscript entitled “Mechanisms of antitumor invasion and metastasis of the ma-rine fungal derivative EAA in HT1080 cells” (Manuscript Number: marinedrugs-2249015).
We have studied the reviewer’s comments carefully and have made revision in the manuscript. Point-by-point responses to the reviewers are listed below this letter. We have tried our best to revise our manuscript according to the comments.
Thank you for your kind comments and careful work.
Authors have presented the Mechanisms of antitumor invasion and metastasis of the marine fungal derivative EAA in HT1080 cells. The concept of MS is good. However, the manuscript can be improved by improving experimental section.
The specific comments, which could help to improve the manuscript are:
Comment 1: Manuscript should be revised for grammatical & punctuation errors.
Response 1: Thanks for your suggestions. We feel sorry for our poor writings. We have been revised in a native English speaker help to polish our article. And we hope the revised manuscript could be acceptable for you.
Comment 2: Better to Provide Full form of EAA in the Title.
Response 2: Thank you for your kind comments and careful work. According to your comments, the related content has been corrected in our revised manuscript.
“Mechanisms of antitumor invasion and metastasis of the ma-rine fungal derivative EAA in HT1080 cells” Corrected to “Mechanisms of antitumor invasion and metastasis of the marine fungal derivative epi-aszonalenin A in HT1080 cells”
Comment 3: Page 7; line 195: How Authors are sure about the stability of complex without performing MD simulation studies?
Response 3: Thank you for your kind comments and careful work. Alkaloids are a group of nitrogen-containing basic organic compounds found in nature, most of which have complex ring structures, with the nitrogen mostly contained within the rings, and are generally stable in nature. They have been extensively studied in recent years and have been found to have a variety of biological activities such as anti-inflammatory, antibacterial, analgesic, antiviral and antitumour activities [1-3]. Vincristine is an alkaloid anti-microtubule drug in clinical use and has been used to treat patients with lymphoma [4]. The metabolic stability and plasma protein binding properties of the alkaloids have also been investigated by researchers and the results obtained are very promising [5]. We have not been able to accurately determine the stability of EAA for the time being, and further in-depth studies will follow to explore its various active effects at this time.
Reference:
[1] Isah T. (2016). Anticancer Alkaloids from Trees: Development into Drugs. Pharmacognosy reviews, 10(20), 90–99.
[2] Cushnie, T. P., Cushnie, B., & Lamb, A. J. (2014). Alkaloids: an overview of their antibacterial, antibiotic-enhancing and antivirulence activities. International journal of antimicrobial agents, 44(5), 377–386.
[3] Bai, R., Yao, C., Zhong, Z., Ge, J., Bai, Z., Ye, X., Xie, T., & Xie, Y. (2021). Discovery of natural anti-inflammatory alkaloids: Potential leads for the drug discovery for the treatment of inflammation. European journal of medicinal chemistry, 213, 113165.
[4] Mateos, J., Pérez-Simón, J. A., Caballero, D., Castilla, C., Lopez, O., Perez, E., Cañizo, C., Vazquez, L., & San Miguel, J. F. (2006). Vincristine is an effective therapeutic approach for transplantation-associated thrombotic microangiopathy. Bone marrow transplantation, 37(3), 337–338.
[5] Obeng, S., Kamble, S. H., Reeves, M. E., Restrepo, L. F., Patel, A., Behnke, M., Chear, N. J., Ramanathan, S., Sharma, A., León, F., Hiranita, T., Avery, B. A., McMahon, L. R., & McCurdy, C. R. (2020). Investigation of the Adrenergic and Opioid Binding Affinities, Metabolic Stability, Plasma Protein Binding Properties, and Functional Effects of Selected Indole-Based Kratom Alkaloids. Journal of medicinal chemistry, 63(1), 433–439.
Comment 4: Page 9, line 269-270: EAA (Figure 1A) was purified from the secondary metabolite of the endophytic fungus A. terreus C23-3 secondary metabolite from coral Pectinia Paeonia. Provide a suitable reference.
Response 4: Thank you for your kind comments and careful work. According to your comments, the related content has been corrected in our revised manuscript.
In 4. Materials and Methods section:
“EAA (Figure 1A) was purified from the secondary metabolite of the endophytic fungus A. terreus C23-3 secondary metabolite from coral Pectinia paeonia and demon-strated its anti-inflammatory and anti-angiogenic activities in our previous study.” Corrected to “EAA (Figure 1A) was purified from the secondary metabolite of the endophytic fungus A. terreus C23-3 secondary metabolite by silica gels vacuum liquid chromatog-raphy (VLC) elution and high performance liquid chromatography from coral Pectinia paeonia, and demonstrated its anti-inflammatory and anti-angiogenic activities in our previous study [11].”
Comment 5: If authors referred earlier related publications, provide a suitable reference for Section 4.2 to 4.7.
Response 5: Thanks for your careful work. In most experiments we followed the reagent instructions. We refer to the experimental methods of western blot and immunofluorescence in earlier papers. According to your comments, the related content has been corrected in our revised manuscript.
In 4. Materials and Methods section:
4.6. Western blot
HT1080 cells were incubated with EAA (0.1, 1 and 10 μM) and PMA (10 ng/mL) in a CO2 incubator for 24 h. Refer to the method of Chen et al. [38]. Proteins were ob-tained by lysing cells using RIPA buffer containing 1% protease inhibitor Phenylme-thylsulfonyl fluoride (PMSF). Separation was performed using SDS-PAGE. Then, they were transferred to nitrocellulose (NC) filter membranes for primary and secondary antibody incubation, and finally, the membranes were photographed using an en-hanced chemiluminescence detection system (Syngene, Cambridge, UK), and the im-ages were analyzed using Image J software.
4.7. Immunocytochemistry
Cells were treated as previously described. Refer to the method of Pei et al. [39]. The cells were fixed with 4% paraformaldehyde for 1 h and permeabilized with 0.2% Triton X-100 for 10 min. Primary and secondary antibodies were incubated after blocking with 5% BSA, and the nuclei were stained with DAPI. Finally, images were recorded by inverted fluorescence microscopy (Olympus, Tokyo, Japan).
In Reference section:
- Chen, J.; Tan, L.; Li, C.; Zhou, C.; Hong, P.; Sun, S.; Qian, Z. J., Mechanism Analysis of a Novel Angiotensin-I-Converting Enzyme Inhibitory Peptide from Isochrysis zhanjiangensis Microalgae for Suppressing Vascular Injury in Human Umbilical Vein Endothelial Cells. J. Agric. Food Chem. 2020, 68, 4411-4423.
- Zheng, H.; Pei, Y.; Zhou, C.; Hong, P.; Qian, Z. J., Amelioration of atherosclerosis in ox-LDL induced HUVEC by sulfated polysaccharides from Gelidium crinale with antihypertensive activity. Int. J. Biol. Macromol. 2023, 228, 671-680.
Comment 6: Page 12; line 356: On what basis authors claim “EAA has the potential to be developed into a functional food?
Response 6: Thank you for your kind comments and careful work. So-called functional foods are a new class of products that promise to improve the targeted physiological functions of the consumer. Due to the increased incidence of lifestyle-related diseases, there is a growing interest in functional foods that are beneficial for bioregulation. Our current research has identified EAA as having activity that interferes with atherosclerosis and tumour development, and if this activity can be applied to some extent, it may be developed as one of the ingredients for functional foods. According to your comments, the related content has been corrected in our revised manuscript.
In 5. Conclusions section:
“In conclusion, EAA has the potential to be developed into a functional food or even a drug that can act effectively in the treatment of angiogenesis-related diseases.” Corrected to “In conclusion, EAA has the potential to be developed into a functional food ingredient or even a drug that can act effectively in the treatment of angiogenesis-related diseases.”
Comment 7: Authors mentioned in Abstract that “EAA had good intervention effects on PMA-induced migration and invasion of HT1080 and reduced the activity of MMPs and vascular endothelial growth factor (VEGF). However, Molecular Docking Studies were performed only on MMP-2/-9.
Response 7: Thank you for your kind comments and careful work. As mentioned in the abstract, "EAA had good intervention effects on PMA-induced migration and invasion of HT1080 and reduced the activity of MMPs and vascular endothelial growth factor (VEGF)." is the result of a Western blot assay, which demonstrated the ability of EAA to regulate MMPs and VEGF after protein assays. In contrast, we chose MMP-2 and MMP-9 for molecular docking because the study focuses on the inhibition of tumour invasion and metastasis by EAA. Both secretory and intracellular proteins can be inhibited by EAA. Molecular docking further confirms the intermolecular interaction of EAA with MMP-2 and MMP-9. According to your comments, the related content has been corrected in our revised manuscript.
In Abstract section:
“The results showed that EAA had good intervention effects on PMA-induced migration and inva-sion of HT1080 and reduced the activity of MMPs and vascular endothelial growth factor (VEGF).” Corrected to“The results of cell wound healing and trans well chamber assay showed that EAA interfered well with PMA-induced migration and invasion of HT1080 cells. Western blot and ELISA assay showed that EAA decreased MMPs and vascular endothelial growth factor (VEGF) activity.”
Comment 8: It would be better to add research gap and future prospects related to the topic in conclusion section.
Response 8: Thank you for your kind comments and careful work. According to your comments, the related content has been corrected in our revised manuscript.
“In conclusion, EAA has the potential to be developed into a functional food ingredient or even a drug that can act effectively in the treatment of angiogenesis-related diseases. Additionally, this study provides further theoretical guidance for the medicinal use of coral-derived endophytes and research data for further experimental studies on EAA.” Corrected to “This study provides further theoretical guidance for the medicinal use of coral endophytes and research data for further experimental studies of EAA. In addition, EAA has the potential to be developed as a functional food ingredient or even a drug that could be effective in treating angiogenesis-related diseases, but further in vivo animal studies and pharmacokinetics are needed to confirm its stability and assess specific tumor suppression effects.”
We would like to express our great appreciation to you for comments on our paper.
Thank you and best regards.
Yours sincerely
Zhong-Ji Qian, Ph.D.
Professor
Southern Marine Science and Engineering Guangdong Laboratory, Zhanjiang, China
Shenzhen Institute of Guangdong Ocean University, Shenzhen 518108, China
School of Chemistry and Environment, Guangdong Ocean University, Zhanjiang 524088, China
Tel: +88-759-2396270
E-mail: zjqian78@163.com

Reviewer 3 Report
This article described about the antitumor mechanisms of EAA.
The experimental design and the results seem to be exquisite and valid.
However, the following point would be considered to improve this article.
Minor point
1) The “epithelial mesenchymal transformation” is described in line 41
for the first time.
Line 41 : epithelial mesenchymal transformation
→ epithelial mesenchymal transformation (EMT)
Line 230 : epithelial mesenchymal transformation (EMT)
→ EMT
2) It is better to describe the abbreviation word “(EAA)” after the description of
full name, because authors have not defined “Epi-aszonalenin A” as “EAA”
before this part except for Abstruct.
Line 68 : EAA → Epi-aszonalenin A (EAA)
3) It is better to describe the abbreviation word “(PMA)” after the description of full name, because authors have not defined “Phorbol-12-myristate-13-
acetate” as “PMA” before this part.
Line 73 : PMA → Phorbol-12-myristate-13-acetate (PMA)
4) The bottom view in “Figure 7”
It is difficult to understand the meaning of the long red arrows.
Dose the long red arrow mean the inhibition of each “signaling pathway” or
“phosphorylation” ?
5) Poor information
a) Line 290-308
How many cells did you seed or used in “Cell activity assay”,
“Cell wound healing assay”, and “Cell invasion by the trans well
chamber assay” ?
b) Line 318
It is better to describe the explanation of “PMSF”.
c) Line 326-327
There are no information of the concentrations or dilution ratios of
primary and secondary antibodies, and conditions, such as reaction
temperature and reaction time.

Author Response
Response to Reviewer 3
Dear editor and reviewers:
On behalf of my co-authors, we thank you very much for giving us an opportunity to revise our manuscript and we appreciate editor and reviewers very much for your positive and constructive comments and suggestions on our manuscript entitled “Mechanisms of antitumor invasion and metastasis of the ma-rine fungal derivative EAA in HT1080 cells” (Manuscript Number: marinedrugs-2249015).
We have studied the reviewer’s comments carefully and have made revision in the manuscript. Point-by-point responses to the reviewers are listed below this letter. We have tried our best to revise our manuscript according to the comments.
Thank you for your kind comments and careful work.
This article described about the antitumor mechanisms of EAA. The experimental design and the results seem to be exquisite and valid. However, the following point would be considered to improve this article.
Minor point
Comment 1: The “epithelial mesenchymal transformation” is described in line 41 for the first time.
Line 41: epithelial mesenchymal transformation
→epithelial mesenchymal transformation (EMT)
Line 230: epithelial mesenchymal transformation (EMT)
→ EMT
Response 1: Thanks for your careful work. According to your comments, the related content has been corrected in our revised manuscript.
In 1. Introduction section:
“Angiogenesis is one of the key factors in tumor metastasis. Tumor metastasis includes epithelial mesenchymal transformation, migration, and invasion of tumors. In-vasion is the first step in tumor metastasis.” Corrected to “Angiogenesis is one of the key factors in tumor metastasis. Tumor metastasis includes epithelial mesenchymal transformation (EMT), migration, and invasion of tumors. In-vasion is the first step in tumor metastasis.”
In 3. Discussion section:
“VEGF promotes the growth of endothelial factors in arterial, venous, and lymphatic vessels. It can specifically affect blood vessels and promote angiogenesis during physi-ological and pathological processes [25]. N-cadherin is a marker of epithelial mesenchymal transformation (EMT) [26]” Corrected to “VEGF promotes the growth of endothelial factors in arterial, venous, and lymphatic vessels. It can specifically affect blood vessels and promote angiogenesis during physi-ological and pathological processes [25]. N-cadherin is a marker of EMT [26]”
Comment 2: It is better to describe the abbreviation word “(EAA)” after the description of full name, because authors have not defined “Epi-aszonalenin A” as “EAA” before this part except for Abstruct.
Line 68: EAA → Epi-aszonalenin A (EAA)
Response 2: Thanks for your careful work. According to your comments, the related content has been corrected in our revised manuscript.
In 1. Introduction section:
“EAA is an alkaloid isolated from a secondary metabolite of the marine coral endophytic fungus Aspergillus terreus C23-3 (Figure 1A).” Corrected to “Epi-aszonalenin A (EAA) is an alkaloid isolated from a secondary metabolite of the marine coral endophytic fungus Aspergillus terreus C23-3 (Figure 1A).”
Comment 3: It is better to describe the abbreviation word “(PMA)” after the description of full name, because authors have not defined “Phorbol-12-myristate-13-acetate” as “PMA” before this part.
Line 73: PMA → Phorbol-12-myristate-13-acetate (PMA)
Response 3: Thanks for your careful work. According to your comments, the related content has been corrected in our revised manuscript.
In 1. Introduction section:
“PMA can act as a pro-cancer factor by activating PKC [12], and therefore is often used as an effective tumor promoter.” Corrected to “Phorbol-12-myristate-13-acetate (PMA) can act as a pro-cancer factor by activating PKC [12], and therefore is often used as an effective tumor promoter.”
Comment 4: The bottom view in “Figure 7” It is difficult to understand the meaning of the long red arrows. Dose the long red arrow mean the inhibition of each “signaling pathway” or “phosphorylation”?
Response 4: Thank you for your kind comments and careful work. The red arrow represents the inhibition of EAA, the phosphorylated protein is indicated by P and the arrow from P downwards is the inhibition of phosphorylation.
Comment 5: Poor information
- a) Line 290-308
How many cells did you seed or used in “Cell activity assay”,
“Cell wound healing assay”, and “Cell invasion by the trans well
chamber assay”?
- b) Line 318
It is better to describe the explanation of “PMSF”.
- c) Line 326-327
There are no information of the concentrations or dilution ratios of
primary and secondary antibodies, and conditions, such as reaction
temperature and reaction time.
Response 5: Thanks for your careful work. According to your comments, the related content has been corrected in our revised manuscript.
In 4. Materials and Methods section:
4.2. Cell activity assay (CCK-8)
HT1080 cells were grown in DMEM with 10% FBS and 1% penicillin/streptomycin. Cells were seeded in 96-well plates (1 × 104 cells/well) and different concentrations of EAA were added to the wells for 24 h. Add 10 μL of Cell Counting Kit-8 (CCK-8) working solution and incubate for 1 h. The absorbance was measured using a microplate reader (BioTek, Winooski, VT, USA) to determine the OD value at 450 nm.
4.3. Cell wound healing assay
HT1080 cells were seeded in 24-well plates (1 × 105 cells/well) and cultured for 24 h at 100% density. Use a sterile tip to scrape the cells. After washing away cellular debris, cells were treated with EAA for 1 h, and then stimulated with PMA (10 ng/mL). The migration of cells across the injury line was observed with a microscope (Olympus, Tokyo, Japan) and photographed at 6 h and 12 h, respectively.
4.4. Cell invasion by the trans well chamber assay
A layer of matrix gel (NEST Biotechnology, Wuxi, China) was placed on the upper layer of the cell compartment of the 24-well plate in advance and left to stand in the incubator for 30 min. Then 200 μL of cell (1 × 105 cells/well) suspension and EAA were added to the cell chambers and incubated in the incubator for 24 h. Finally, cells were stained with crystalline violet, observed with a microscope (Olympus, Tokyo, Japan), and photographed.
4.6. Western blot
HT1080 cells were incubated with EAA (0.1, 1 and 10 μM) and PMA (10 ng/mL) in a CO2 incubator for 24 h. Refer to the method of Chen et al. [38]. Proteins were obtained by lysing cells using RIPA buffer containing 1% protease inhibitor Phenylmethylsulfonyl fluoride (PMSF). Separation was performed using SDS-PAGE. The proteins were then transferred onto nitrocellulose (NC) filter membranes. Primary antibodies were incubated overnight at 4°C (1:1000), followed by sufficient incubation of secondary antibodies for 2 h (1:2000), and finally, the membranes were photographed using an enhanced chemiluminescence detection system (Syngene, Cambridge, UK), and the images were analyzed using Image J software.
We would like to express our great appreciation to you for comments on our paper.
Thank you and best regards.
Yours sincerely
Zhong-Ji Qian, Ph.D.
Professor
Southern Marine Science and Engineering Guangdong Laboratory, Zhanjiang, China
Shenzhen Institute of Guangdong Ocean University, Shenzhen 518108, China
School of Chemistry and Environment, Guangdong Ocean University, Zhanjiang 524088, China
Tel: +88-759-2396270
E-mail: zjqian78@163.com

Round 2
Reviewer 1 Report
Thanks for your corrections.